# Stem Cell Therapy for Acute/Subacute Ischemic Stroke with a Focus on Intraarterial Stem Cell Transplantation: From Basic Research to Clinical Trials

**DOI:** 10.3390/bioengineering10010033

**Published:** 2022-12-27

**Authors:** Susumu Yamaguchi, Michiharu Yoshida, Nobutaka Horie, Katsuya Satoh, Yuutaka Fukuda, Shunsuke Ishizaka, Koki Ogawa, Yoichi Morofuji, Takeshi Hiu, Tsuyoshi Izumo, Shigeru Kawakami, Noriyuki Nishida, Takayuki Matsuo

**Affiliations:** 1Department of Neurosurgery, Graduate School of Biomedical Sciences, Nagasaki University, Nagasaki 852-8501, Japan; 2Department of Neurosurgery, Sasebo General Hospital, Nagasaki 857-8511, Japan; 3Department of Neurosurgery, Hiroshima University, Hiroshima 734-8551, Japan; 4Department of Occupational Therapy Sciences, Graduate School of Biomedical Sciences, Nagasaki University, Nagasaki 852-8501, Japan; 5Department of Pharmaceutical Informatics, Graduate School of Biomedical Sciences, Nagasaki University, Nagasaki 852-8588, Japan; 6Department of Molecular Microbiology and Immunology, Graduate School of Biomedical Sciences, Nagasaki University, Nagasaki 852-8523, Japan

**Keywords:** intraarterial transplantation, regenerative medicine, stem cell transplantation, ischemic stroke

## Abstract

Stem cell therapy for ischemic stroke holds great promise for the treatment of neurological impairment and has moved from the laboratory into early clinical trials. The mechanism of action of stem cell therapy includes the bystander effect and cell replacement. The bystander effect plays an important role in the acute to subacute phase, and cell replacement plays an important role in the subacute to chronic phase. Intraarterial (IA) transplantation is less invasive than intraparenchymal transplantation and can provide more cells in the affected brain region than intravenous transplantation. However, transplanted cell migration was reported to be insufficient, and few transplanted cells were retained in the brain for an extended period. Therefore, the bystander effect was considered the main mechanism of action of IA stem cell transplantation. In most clinical trials, IA transplantation was performed during the acute and subacute phases. Although clinical trials of IA transplantation demonstrated safety, they did not demonstrate satisfactory efficacy in improving patient outcomes. To increase efficacy, increased migration of transplanted cells and production of long surviving and effective stem cells would be crucial. Given the lack of knowledge on this subject, we review and summarize the mechanisms of action of transplanted stem cells and recent advancements in preclinical and clinical studies to provide information and guidance for further advancement of acute/subacute phase IA stem cell transplantation therapy for ischemic stroke.

## 1. Introduction

Recent advancements in treatments for ischemic stroke have been remarkable. Interventions such as intravenous administration of recombinant tissue plasminogen activator and mechanical thrombectomy have improved the outcomes of patients with ischemic stroke [1]. These treatments focus on reperfusion in the hyperacute phase, thereby excluding patients admitted to the hospital in other phases of the disease. Once the neurological impairments have progressed, these patients have no options other than rehabilitation. Stem cell therapy holds great promise for treating neurological impairment after ischemic stroke. Basic research on stem cell therapy has revealed the multipotent effects of transplanted stem cells, including the secretion of various growth factors. Recent advancements in genetic engineering enable the production of gene-modified stem cells to improve therapeutic efficacy [2]. The clinical induction of new drug delivery systems, such as those used for coronavirus disease vaccines, may provide patients who have experienced stroke with new therapeutic options [3]. mRNA induction in the brain has been achieved using mRNA encapsulated in lipid nanoparticles (LNPs) [4]. Theoretically, this technique, combined with endovascular techniques and/or a focused ultrasound (FUS)/microbubble-assisted blood–brain barrier (BBB) opening technique, would allow mRNA to induce all growth factors in the targeted brain locations at the correct time. Although the analysis of the gene expression of transplanted stem cells in basic research has advanced gene therapy, not all underlying processes and effects have been revealed. As the role of transplanted stem cells is further elucidated, new types of drug therapy, such as mRNA encapsulated in LNPs, could be used to pharmaceutically treat ischemic stroke.

Stem cell therapy has moved from the laboratory into early clinical trials [5]. Although many clinical trials have demonstrated the safety of stem cell transplantation for ischemic stroke, only some trials were able to prove evidence of their therapeutic efficacy [6]. In a recent randomized controlled trial (RCT) published in 2019, which used IA transplantation for cerebral infarction, researchers failed to show therapeutic efficacy; however, they did demonstrate safety as there were no occurrences of symptomatic embolic stroke, which is one of the adverse effects of IA transplantation requiring attention [7]. The next focus should be on improving therapeutic efficacy without increasing adverse effects. In this article, we review the pathology of ischemic stroke and the mechanism of action of stem cell therapy in ischemic stroke, with a focus on acute and subacute phase stem cell IA transplantation. We discuss the current challenges and future directions from the perspective of our findings.

## 2. Mechanisms of Action of Stem Cell Transplantation in Ischemic Stroke

Stem cell therapy action is based on two mechanisms: the bystander effect and cell replacement. In the bystander effect, the transplanted stem cells secrete various paracrine factors to modulate the host environment, while cell replacement involves the differentiation of stem cells into various cell lineages, which then participate in remodeling.

## 3. Features of IA Stem Cell Transplantation

Stem cell transplantation has two routes of administration: systemic and local. Systemic administration includes intravenous (IV) and IA transplantation, while local administration involves intracerebral (IC) and intracerebroventricular administration. In a study using a rodent model, stem cells were systemically administered by IA transplantation via the cervical carotid artery [8,9,10,11]. Other studies on IV transplantation used the femoral or tail veins as the method of approach [12,13]. Studies on IC transplantation used an approach of stereotaxically transplanting stem cells into the periinfarct area [14,15]. Stem cells administered systemically have been shown to better modulate systemic immune responses over those administered locally [16]. Local administration, especially IC transplantation, may only modulate the local immune response. However, it allows more cells to be placed into the brain, leading to a higher replacement effect over that of systemic administration. IA transplantation is more advantageous than IV transplantation since it allows selective distribution of a larger number of cells into the target lesion [17,18]. Although fewer cells are placed in the brain with IA transplantation than with IC transplantation, IA transplantation is less invasive and can provide an effective bystander effect: it does not induce brain injury or result in placement of transplanted stem cells in the spleen [19,20]. IV transplantation may be considered superior to IA transplantation in terms of the peripheral immunomodulatory effect of transplanted stem cells, as more transplanted stem cells are trapped in the spleen with IV than with IA transplantation [19].

The dose of the transplanted stem cells plays a crucial role in the outcome of the transplant. IA transplantation can incorporate many stem cells into the infarcted region of the brain. BBB dysfunction, especially in acute phase transplantation, can facilitate the migration of larger numbers of transplanted cells into the brain. However, as the volume of infused stem cells increases, the risk of embolization also increases, which could further aggravate the infarction. Although the preservation of antegrade flow reportedly prevents microembolic stroke [21], high-dose IA transplantation is associated with an increased rate of embolism and high mortality [9,10]. Thus, the optimal dose of transplanted stem cells needs to be determined.

The timing of transplantation affects the role it plays. The aim of acute phase stem cell transplantation is neuroprotection, while that of chronic phase stem cell transplantation is to remodel the damaged brain [22]. One study showed that the timing of IV transplantation in the acute and chronic phases did not affect functional recovery [22]. However, we demonstrated that acute phase IA transplantation showed better functional recovery than subacute transplantation [8]. Following an IA transplantation performed 1 day after transient middle cerebral artery occlusion (MCAO), the transplanted mesenchymal stem cells (MSCs) were uniformly distributed in both the infarct core and periphery 72 h after the procedure. In IA MSC transplantation 4 days after ischemic stroke, the number of transplanted MSCs in the infarct core was significantly lower than that with transplantation 1 day after ischemic stroke. Similarly, IA MSC transplantation 7 days after ischemic stroke showed a further decrease in transplanted MSCs in both the infarct core and periinfarct lesion [8]. Thus, in IA stem cell transplantation, the timing of transplantation can affect the distribution of transplanted stem cells, affecting the host environment in the recipient brain.

## 4. Cell Source/Cell Quality

Stem cells from various sources have been used in transplantation. Although there was a comparative study comparing different cell sources [23], the most suitable cell source for transplantation has not been confirmed. Since there is transdifferentiation of transplanted stem cells [23], and stem cells have multiple therapeutic effects, the cell source did not appear to be selected by the specific treatment target, e.g., only neurogenesis or only angiogenesis. Owing to the easier and faster process of collection and isolation of bone marrow mononuclear cells (BMMCs), less preparation time is required for their use as stem cells, and BMNCs have been used in many acute/subacute phase autologous transplantation studies [7,19,24,25,26,27,28]. BMMCs primarily cause angiogenesis; they can increase vascular density and blood flow [2]. However, BMMCs are not a sufficiently rich source of stem cells; therefore, their neurogenesis or replacement effect would be limited. On the contrary, MSCs can be sourced from both the bone marrow and adipose tissue. Although the risk of tumorigenicity of MSCs was inconclusive [29], MSCs have immunotolerance [30] and have been widely used to treat various diseases [31]; therefore, MSCs were used in many studies. MSCs have multipotent effects. They secrete various cytokines/growth factors, which can cause angiogenesis, antiinflammation, and neurogenesis. MSCs can transdifferentiate into endothelial cells and neurons [2,8,15]. Although neural stem cells (NSCs) are traditionally expected to replace lost neurons and glia cell, owing to their potential for differentiating into neural cells, NSCs were recently found to have multiple additional properties, including prevention of neuronal apoptosis, angiogenesis, endogenous neurogenesis, and immunomodulation by secretion of various cytokines/growth factors [2]. However, the application of embryonic stem (ES) cells derived from NSCs in clinical studies raises ethical concerns. It is noteworthy that dental pulp stem cells (DPSCs) reside within the perivascular niches of dental pulp, can be easily collected, and have the features of both NSCs and MSCs; DPSCs can enhance angiogenesis, neurogenesis, and antiinflammation by secretion of various cytokines/growth factors. Although for use of DPSCs as stem cells, longer cell preparation time is required compared with BMMCs, DPSCs also have an immunotolerance function; therefore, DPSCs may be useful for allogeneic transplantation [2]. Induced pluripotent stem (iPS) cells can differentiate into all types of cells, and there are no associated ethical issues as opposed to ES cells. iPS cells are also suitable for autologous transplantation if they are produced by the patients’ own cells. Therefore, iPS cells are attractive as a cell source for transplantation. However, iPS cells involve a risk of tumorigenicity and the cell preparation process is lengthy [2]. Further improvement in extraction technique is necessary.

The quality of stem cells also affects functional recovery after transplantation. Advancing age, sex, history of smoking, obesity, and presence of comorbidities can possibly affect the quality of stem cells. Aging is a key factor decreasing stem cell quality. MSCs derived from the elderly have shown decreased secretion of brain-derived neurotrophic factor (BDNF) and platelet-derived growth factor-BB (PDGF-BB) [11]. MSCs showed decreased functionality, with reduced angiogenesis and neurogenesis, when they were derived from older rather than younger people [11]. Differences in stem cell function are affected by sex; a study demonstrated that differences in gene expression of human adipose tissue-derived MSCs were associated with sex [32]. MSCs from patients with type 1 diabetes mellitus have been shown to secrete degenerated exosomes [33]. Additionally, it was reported that MSCs from obese people could not decrease the M1/M2 ratio of macrophages in renal artery stenosis and thus demonstrated impaired anti-inflammatory effects [34]. Smoking was also reported to decrease MSC function, such as proliferation, paracrine effect, and differentiation [35]. Thus, it is postulated that comorbidities and lifestyle factors, such as smoking and obesity, may also contribute to decreased stem cell functionality, especially in autologous transplantation.

## 5. Bystander Effects

### 5.1. Angiogenesis

Vascular endothelial growth factor (VEGF) and PDGF-BB play vital roles in angiogenesis. VEGF enhances endothelial cell (EC) proliferation and increases the permeability of the vessels [36]. PDGF-BB enhances pericyte proliferation [37]. One to two hours after ischemic stroke, the cerebral microvessels of the infarct core begin to overexpress VEGF and integrin αv subunits [38]. Although gene expression of VEGF was not detected 3 h post ischemic stroke, expression greatly increased 6 h after occlusion, peak expression occurred on day 3, and expression returned to normal at the border of the infarct area 7 days post stroke. VEGF receptors (VEGFRs) on microvessels in the periinfarct area strongly expressed, peaked, and returned to normal by days 2, 3, and 7 post ischemic stroke, respectively. The expression of VEGFR on microvessels started at the border of the infarct area and spread to the infarct core [39]. In addition to the ECs, the neurons, astrocytes, and microglia also secrete VEGF [40,41]. The proliferation of ECs at the border of the ischemic core was observed up to 7 days post ischemic stroke, and the formation of new blood vessels was observed at the border of the infarct core and periinfarct area [41,42]. Angiopoietin-1 (Ang-1) is also crucial for capillary tubule formation [43] and its mRNA expression persists for 28 days post ischemic stroke [44]. After the formation of capillaries, VEGF enhances the coverage of ECs by pericytes 21 days post ischemic stroke [45]. PDGF-BB, derived from ECs, enhanced pericyte coverage for 3 to 7 days post ischemic stroke [46]. This vessel maturation enhances BBB integrity and contributes to decreased neuroinflammation [47].

Pericytes also enhance angiogenesis. Pericytes control the activity of ECs [48]. After ischemic stroke, in response to the stimulation of pro-angiogenic factors, pericytes produce matrix metalloproteinases to degrade the basement membrane, which leads to the release of ECs from the basement membrane and increases vascular permeability [49]. Increased vascular permeability allows plasma protein extravasation, contributing to the formation of a scaffold to sprout the vessels [50]. Pericyte-secreted VEGF also enhances the ECs’ sprouting proliferation [51]. Pericytes attach to ECs, control the deposition of extracellular matrix, and enhance tight junction formation between each EC. Once sufficient vessel sprouting is achieved, pericytes inhibit EC activity to terminate angiogenesis [48].

Thus, to enhance angiogenesis, the opportune time for transplantation would be during the acute or subacute phase. In subacute phase IC transplantation, human VEGF secreted by transplanted stem cells reportedly enhanced angiogenesis, leading to functional recovery in rats [14]. We previously reported that MSC secreted PDGF-BB, and acute phase IA MSC transplantation enhanced mature angiogenesis, depending on the amount of secreted PDGF-BB [11]. Furthermore, the timing of transplantation might affect angiogenesis via transplanted stem cell distribution. As mentioned above, angiogenesis started at the border of the infarct core in the acute phase. In IA transplantation, 1 day after ischemic stroke, transplanted stem cells were distributed to both the peri-infarct lesion and the infarct core [8]. When IA transplantation was performed in the acute phase, the transplanted stem cells at the border of the infarct core caused moderate increase in angiogenesis; this increase in the level of angiogenesis was comparatively higher in the acute phase than that achieved with transplantation in the subacute and chronic phases. IA transplantation performed in the acute phase resulted in a higher degree of infarct volume reduction and function recovery than IA transplantation performed in the subacute phase (summarized in Figure 1) [8].

### 5.2. Neurogenesis

Neurogenesis has been widely studied since the discovery of NSCs from the adult mammalian central nervous system [52]. BDNF, insulin like growth factor-1, fibroblast growth factor-2 (FGF-2), epidermal growth factor (EGF), erythropoietin, and mechano-growth factor have been reported to enhance NSC proliferation [53,54,55,56,57,58]. Ischemic stroke enhances NSC proliferation in the subventricular zone (SVZ), CA1 region, and subgranular zone (SGZ) of the dentate gyrus in the hippocampus, after which the NSCs migrate into the ischemic regions [59]. After ischemic stroke, astrocytes express stromal cell-derived factor-1 (SDF-1) alpha [60], which mediates stem cell migration [61]. NSCs were reported to migrate into the periinfarct cortex via the rostral migratory stream (RMS) [62]. Neural progenitor cells (NPCs) migrate along blood vessels and then along astrocytes from day 3 to day 15, and from day 7 to day 15 post ischemic stroke, respectively [63]. In addition to increasing the number of NSCs [59], BDNF plays an important role in NSC migration into the injured site. The receptor p75NTR on the NSC surface binds to BDNF on the blood vessels and reactive astrocytes [64,65]. In the striatum, neuroblasts expressing Slit1 could control the morphology of reactive astrocytes through the Slit1-binding protein Robo2, and the Slit-Robo signaling pathway, and migrate into the ischemic lesion through the tunnel of reactive astrocytes [66]. Astrocytes and their morphology affect the migration of NSCs.

In addition to NSCs supplied from the SVZ, CA1, and SGZ, via the RMS, the NSCs derived from pericytes contribute to neurogenesis after ischemic stroke. Recently, pericytes were reported to form multipotent stem cells after stroke. After ischemic stroke, pial cells generate Nestin-positive cells in the pia mater at the site of the infarct [67]. These cells, termed ischemic pericytes (iPCs), form neurosphere-like cell clusters in response to basic FGF and EGF in vitro, and have neural stem/progenitor cell activity. These cells differentiate into Tuj1-positive neurons with the neural-conditioned medium [68,69]. The number of iPCs peak by day 3 post ischemic stroke, before decreasing until day 7 [69].

We previously reported that MSCs secrete BDNF and PDGF-BB [11]. IA transplantation of MSCs 1 day post stroke increased BDNF levels in the periinfarct area 7 days post ischemic stroke and enhanced NSC migration by improving the alignment of astrocytes [11]. Moreover, PDGF receptor-beta, on NSCs, which is a PDGF-BB receptor, has been reported to be associated with NSC migration post ischemic stroke [70]. Thus, acute phase stem cell transplantation is considered to modulate the host environment and enhance NSC migration.

A previous report demonstrated that MCAO did not cause proliferation of NSCs in the contralateral SVZ [71]. Musashi-1 (Msi-1) is one marker of NSCs. Our results showed that the Msi-1 positive region in the contralateral SVZ in the control group did not increase after MCAO, consistent with the previous report [71]. However, when IA transplantation was carried out 1 day post MCAO, the Msi-1 positive region of the contralateral SVZ was significantly larger in the treatment than that in the control group when evaluated 2 days post MCAO (Figure 2a). Seven days post MCAO, the number of Msi-1-positive cells on the corpus callosum (CC) was significantly higher in the treatment group than that in the control group (Figure 2b). We previously reported that 21 days post ischemic stroke, the number of Msi-1-positive and glial fibrillary acidic protein-negative cells in the periinfarct area was significantly larger in the treatment group than that in the control group [11]. Considering that activated NSCs migrate into the injury site via the RMS from the SVZ, CA-1, and SGZ after ischemic stroke [62], activated NSCs introduced in the contralateral SVZ by IA transplantation may migrate into the periinfarct area through the CC. Indeed, after IA transplantation, infused stem cells were observed in the contralateral brain [72]. Transplanted MSCs in the contralateral brain and/or the bystander effect of transplanted MSCs may enhance neurogenesis via the RMS in the contralateral brain. However, further studies are needed to confirm this (summarized in Figure 3).

### 5.3. Anti-Inflammatory Processes

A number of anti-inflammatory processes occur in the innate immune system. After tissue damage, damage-associated molecular patterns (DAMPs) induce inflammation. After ischemic stroke, high mobility box 1 (HMGB1), peroxiredoxin (PRX), and heat shock proteins (HSP)—which are included in the DAMPs—induce neuroinflammation.

After ischemic stroke, HMGB1 induces BBB degradation and increases the permeability of the BBB in the hyperacute phase [73]. This increased vessel permeability permits various inflammatory cells to invade the brain. Twelve hours post ischemic stroke, PRX family proteins are released extracellularly from the damaged cells in the infarct core. This activates macrophages and neutrophils via activation of toll-like receptors (TLR) 2 or 4 through inflammatory cytokines, including interleukin (IL)-23, to induce the production of various pro-inflammatory cytokines [74,75]. HSP 60, released by damaged cells, also activates TLR 4, thereby activating microglia [76]. These macrophages/microglia produce IL-1β, which damages neurons [77,78]. There are two types of macrophages/microglia: M1 and M2. M1 macrophages/microglia are pro-inflammatory and have cytotoxic and phagocytic effects that damage neurons. M2 macrophages/microglia are anti-inflammatory and scavenge the debris and secrete trophic factors, leading to tissue remodeling and growth stimulation [79]. Although it is challenging to distinguish between M1 and M2 macrophages/microglia during ischemic stroke, the time course of M1/M2 polarization from onset to 7 days post ischemic stroke has been studied. M1-like macrophages/microglia continue to gradually increase until day 7 post ischemic stroke. The M2-like macrophages/microglia peaked on day 1 post ischemic stroke. Then, the M2-like macrophages/microglia decreased until day 7 post ischemic stroke [80], a trend observed until day 14 post stroke [81]. From day 3 post ischemic stroke, MSR1-expressing macrophages increased, scavenging HMGB1 and PRX until 1 week post ischemic stroke [82]. These events are considered to be involved in decreasing the permeability of the BBB.

DAMPs inducing neuroinflammation also induce delayed inflammation, mediated by T cells, and delayed neuronal injury in the penumbra [83]. T cells were observed in the pial lesion of the ischemic brain on day 1 post ischemic stroke, increased in number, and were observed in the periinfarct lesion by day 7 post ischemic stroke [84,85]. Although the number of T cells decreased by day 14 post ischemic stroke, they were still observed up to one month later [85,86]. CD8+ T cells produce cytotoxic T cells; interferon gamma (IFN-γ), which causes direct neuronal damage; and IL-16, which increases pro-inflammatory cytokines, resulting in recruitment and activation of the immune cells, leading to degradation of the BBB [87]. The peak of IL-16 accumulation in the infarct core is 3–4 days post ischemic stroke [88]. CD4+ T cells are divided into helper T cells (Th) and regulatory T cells (Treg). Th1 is responsible for a delayed immune response to ischemic stroke. The level of IFN-γ is increased in the infarcted brain 72 h post ischemic stroke [16]. IFN-γ attracts Th1 to the infarct site and produces IFN-γ, leading to cytotoxic inflammation [87]. In contrast, Th2 causes a humoral immune response and releases IL-10, which is an anti-inflammatory cytokine and reduces the infarct volume [89]. The number of Tregs continues to gradually increase from the acute phase to 60 days post ischemic stroke [90]. IL-2, IL-33, serotonin, and T cell recognition increase the number of Tregs [90]. Treg also produces IL-10 and contributes to reduction of the infarct volume 7 days post ischemic stroke [91]. In the subacute and chronic phases, Treg produces amphiregulin, a low-affinity EGF receptor ligand, to prevent neurotoxic astrogliosis [90].

Thus, the goal of treatment of ischemic stroke is to inhibit cytotoxic inflammation in the acute phase and to enhance remodeling in the subacute and chronic phases. In an in vitro assessment, MSCs promoted the polarization of macrophages from M1 to M2, reduced the proportion of CD8+ T cells, and increased the proportion of Tregs among T cells [92,93]. We previously reported that IA MSC transplantation performed 1 day after MCAO reduced activated macrophage/microglia infiltration in the periinfarct area 7 days after MCAO [10]. Intravenous MSC transplantation 3 h post ischemic stroke reduced the infiltration of macrophages and increased the infiltration of Tregs in the hemisphere of infarction 3 days post the procedure [93]. Stem cell transplantation in the acute phase was reported to improve BBB integrity [22]; this would prevent inflammatory infiltration. Intraventricular MCS transplantation, performed 1 day post ischemic stroke, reportedly upregulated IL-10 and reduced tumor necrosis factor (TNF)-alpha 4 days post ischemic stroke [94]. IA MSC transplantation 1 day post ischemic stroke also reduced IL-1β 8 days post stroke [10]. IV MSC transplantation 3 h post ischemic stroke reduced INF-γ 24 h post ischemic stroke [93]. Thus, MSC transplantation in the acute phase has a neuroprotective effect by reducing pro-inflammatory cytokines. Furthermore, we also reported that IA MSC transplantation performed 1 day post ischemic stroke increased the MCP-1 level in the periinfarct area 7 days post ischemic stroke [11]. This may contribute to NSC migration and neurite outgrowth in the subacute phase [95,96] (summarized in Figure 4).

### 5.4. Enhancement of the Bystander Effect by Gene Modification

The main role of stem cell therapy in the acute and subacute phases is the bystander effect; however, the bystander effect expressed by gene-modified stem cells is greater. Ang-1 also plays a crucial role in angiogenesis. Regarding vessel maturation, acute phase Ang-1 gene-modified MSC transplantation demonstrated better angiogenesis and functional improvement than non-gene-modified MSC transplantation [97]. Notably, a combined Ang-1 and VEGF gene-modified MSC showed greater treatment efficacy in acute phase transplantation [98]. In neurogenesis, glial cell line-derived neurotrophic factor (GDNF) enhanced the survival and morphological differentiation of motor neurons, while subacute phase BDNF and GDNF gene-modified MSC transplantation reduced the infarct volume and enhanced functional recovery over non-gene-modified MSC transplantation/control [99,100,101]. In anti-inflammatory processes, hepatocyte growth factor (HGF) protected gap junction proteins, preventing BBB injury, while hyperacute phase HGF gene-modified dental pulp stem cells reduced the infarct volume and improved functional recovery over the control condition and use of non-gene modified dental pulp stem cells [102]. Hyperacute phase IL-10 gene-modified MSC transplantation also reduced the infarct volume and improved functional recovery [13]. Combining these gene-modified stem cells and transplantation at an optimal time may improve therapeutic efficacy.

### 5.5. Additional Issues Related to the Bystander Effect

Inflammation controlled by stem cell transplantation increases cell migration—including that of ECs, pericytes, and neural stem cells—resulting in enhanced angiogenesis and neurogenesis. Mature angiogenesis enhances neural stem cell migration to the injury site and BBB integrity, preventing harmful inflammatory cells from infiltrating the injury site. Transplanted stem cells can secrete various growth factors, trophic factors, and cytokines, and these factors synergistically contribute to enhancement of tissue repair, including reconstruction of the neurovascular unit [103].

The systemic effect of systemic stem cell transplantation, including IA transplantation, is mainly dependent on the transplanted stem cells’ ability to modulate peripheral immunity. Transplanted stem cells labeled with technetium-99 were shown to be trapped to a greater extent by the liver than by the spleen in IA transplantation [19], and some transplanted stem cells were found to function in the spleen. In hyperacute phase IV transplantation, the level of CD4+ Th17 cells were reduced and the level of Tregs was increased in the spleen and blood 72 h post ischemic stroke. The levels of IL-1, TNF-alfa, IL-23, and IL-17 in the blood were reduced, and the level of IL-10 was increased until 1 week post ischemic stroke [20]. As the prevention of early infiltration of neutrophils into the brain was reported to be important in hypoxic ischemic brain injury [104], we suggest that modulation of peripheral immunity caused by hyperacute/acute phase stem cell transplantation may be beneficial. However, the effects of long-term peripheral immunomodulation after acute phase stem cell transplantation are not well-known. Furthermore, it is difficult to track transplanted stem cells in the long term. Radioisotopes have a short half-life, and therefore, it is difficult to track them in the long term [105]. Superparamagnetic iron oxide (SPIO)-labeled stem cells can be tracked via magnetic resonance imaging (MRI), thereby enabling long-term tracking of transplanted stem cells [106]. However, in this case, it is difficult to distinguish between surviving transplanted stem cells and cells that have phagocyted apoptotic transplanted stem cells. Thus, it is difficult to evaluate the duration for which transplanted stem cells survive in the human body and for which they exert the bystander effect.

## 6. Cell Replacement

To obtain better efficacy with cell replacement, numerous transplanted stem cells must be placed in the brain, and the transplanted cells should survive for a long time. As the number of transplanted stem cells placed via IA or IV transplantation in the brain were lower than that via IC transplantation, all studies that assessed replacement effects were almost always done by IC transplantation [107]. To evaluate the extent of cell replacement, tracking of transplanted stem cells is required. Transplantation of human stem cells into rat brain is a useful method because it makes it possible to distinguish transplanted cells from host cells. In our study, an IA transplantation was performed 1 day after transient MCAO, and although transplanted human cells were observed 7 days after transient MCAO, few transplanted human cells could be observed 21 days after transient MCAO [11]. In the acute phase post ischemic stroke, the environment of the host’s brain is harmful to the transplanted stem cells because of severe inflammation [108]. Based on the aforementioned findings, it can be concluded that the number of transplanted stem cells that survive in the hyper- and acute phases decreases [8,11,13,102]. Thus, the replacement effect obtained by IA transplantation in the acute and subacute phases appears to be limited. A study performed IV transplantation 1 day after transient MCAO in rats that were treated/not treated with a free radical scavenger and found that on day 14 post MCAO, owing to free radical scavenger—a neuroprotective agent—there was improvement in the host environment, which contributed to improved survival of transplanted stem cells in rats treated with the free radical scavenger compared with rats that were not treated with it; however, stem cells were not assessed for survival beyond the 14 days post ischemic stroke [109].

Although very few stem cells transplanted in the acute phase were observed in the injured brain in the subacute or chronic phase [8,11], it has been noted that transplanted stem cells differentiated into neurons and ECs in acute phase transplantation [8,22]. Multilineage-differentiating stress-enduring (MUSE) cells comprise a type of MSCs [110], and IC transplantation of MUSE cells into the striatum in the subacute phase showed transdifferentiation of MUSE cells into neurons and their involvement in the reconstruction of the pyramidal tract [15]. This indicates that even if transplantation occurs in the subacute phase, if there is persistent neuroinflammation, the placement of a large volume of stem cells in the brain could provide an effective replacement effect. Thus, further studies focusing on increasing stem cell placement and creating long-surviving stem cells are needed to improve the replacement effect of IA transplantation in both the acute and subacute phases.

Recently, a new mechanism of action of replacement by stem cells has been reported. Although stem cells that migrate into ischemic lesions transdifferentiate into various cell lineages by the paracrine effect, MUSE cells, in fact, phagocytose the apoptotic differentiated cells and thereby become differentiated [111]. Thus, stem cells replace the differentiated cells by the paracrine effect or by phagocytosing apoptotic differentiated cells.

## 7. Clinical Trials of IA Stem Cell Transplantation in the Acute and Subacute Phases

Nine clinical studies of IA transplantation after cerebral infarction were reported (Table 1) [7,19,24,25,26,27,28,112,113]. IA transplantation was either performed at approximately, or within, 1 week (acute phase), between approximately 1 week and 1 month (subacute phase), and after 1 month (chronic phase) post cerebral infarction, in three, five, and one study, respectively. Many studies used autologous BMMCs. The cells were administered using endovascular approaches via the middle-cerebral and internal-carotid arteries in seven and two studies, respectively. Adverse events related to endovascular treatment, such as hemorrhages at the puncture site, allergy to contrast medium, renal impairment, arterial dissection, and cerebral infarction, are more specific to IA transplantation compared with IV and IC transplantations [114]. In one of these studies, although 4 out of 29 patients had asymptomatic infarction evaluated by MRI, there were no severe procedure-related adverse events despite the maximal cell dose of 6 × 10^8^, maximum cell density of 5 × 10^7^/mL, and maximum infusion speed of 1.75 mL/min. Although the adverse event that should be given the most attention remains embolic stroke, seizure also seemed to be a noteworthy adverse event related to IA transplantation. In all the studies, the cerebral infarction was an MCA lesion, and there were no patients with cerebral infarction in the posterior circulation nor in only the anterior cerebral artery circulation. Four of the nine studies were comparative studies involving control groups, and two of these studies were RCTs [7,19,27,28]. In one of the four studies (one of the two RCTs), there were signs of more favorable outcomes, of neurological recovery, in the cell transplantation group than in the control groups [28], while others did not show significant cell transplantation efficacy. In all studies where transplantation occurred within 1 month of cerebral infarction, patients receiving transplantation showed neurological improvement. However, since it is difficult to distinguish the transplantation effect from the natural course/spontaneous recovery after ischemic stroke, clinical studies could not determine the efficacy of IA transplantation in the acute/subacute phase.

Four studies without a control group and three RCTs have examined IV transplantation in the acute and subacute phases [115,116,117,118,119,120,121]. The cells were administered via peripheral veins including the antecubital vein. In all the studies, the authors concluded that there were no adverse events related to transplantation. No RCT could show significant treatment efficacy differences between the treatment and control groups [117,119,121]. One study reported that patients who received high-dose cell transplantation (mean 3.4 × 10^7^) tended to have better functional recovery than those who received low doses (2.5 × 10^7^) [118]. Since only one RCT of IA transplantation demonstrated considerable treatment efficacy, the fact that IA transplantation could place more stem cells than IV likely contributed to treatment efficacy in that trial. Concerning adverse events, IA transplantation was associated with adverse effects that were not deadly.

## 8. Future Directions

Cell transplantation therapy for ischemic stroke has moved from the laboratory into early clinical trials, which have demonstrated the safety of IA transplantation in the acute and subacute phases of ischemic stroke. However, its efficacy has been reported to be unsatisfactory. Problems encountered in ensuring the efficacy of stem cell transplantation therapy include the proper selection of transplanted cells and the cell dose, and these have been elucidated by discrepancies in the findings of the clinical studies and the animal-model studies.

Consequently, as aforementioned, many studies have utilized stem cells with multipotential bystander effects, and there have been a plethora of clinical trials on BMMCs and their primary role in angiogenesis [2]. A study reported that a better Barthel index 1 month after transplantation correlated with a higher number of CD34+ cells [26]. Since BMMCs contain a small number of hematopoietic stem cells and MSCs, the bystander effect would therefore be limited [2,19]. The preparation of MSCs—from bone marrow aspiration to transplantation—is time consuming [2]. This prevents acute and subacute phase transplants from using autologous MSCs. Although stock of patients’ own MSCs, or allogenic transplantation, could resolve this issue in MSC transplantation in the acute and subacute phases, there have been no reports of clinical trials using stocks of patients’ own MSCs or allogenic MSCs in IA transplantation. In clinical trials, cell dose was determined by referring to the results of preclinical studies using animal models and included a safety margin to avoid embolic stroke caused by stem cell transplantation. Thus, in clinical trials, the cell dose may not have been sufficient to demonstrate treatment efficacy.

Factors that limit the robustness of preclinical research include the use of animal models and the non-publication rates and publication bias in animal research. Most patients with cerebral infarction are elderly, and in those patients, drawbacks such as stem cell aging or decreased stem cell function can possibly interfere with efficacy [11]. Indeed, preclinical animal model studies use healthy, young, male adult animals—which are not representative of humans in the real world, especially since individuals who require cell transplantation tend to be older and have various comorbidities [108]. Thus, preclinical studies might not sufficiently reflect the real-world setting. Notably, rodent models have also been reported to show spontaneous recovery, which makes the evaluation of the efficacy of cell transplantation challenging [122]. Although many preclinical studies have demonstrated the efficacy of cell transplantation in vivo using animal models, Bliss et al. pointed out that many preclinical studies using animal models that obtained negative data and/or results have not been published [108]. Therefore, the causes of failure in such studies have not been sufficiently assessed. Thus, further examination would be required to address the discrepancy.

To increase therapeutic efficacy, stem cell functionality and migration to the infarct site need to be enhanced.

### 8.1. Enhancement of Stem Cell Functionality

Although the details of gene-modified stem cells have been stated above, various stem cell preconditioning strategies have also been reported. Since hypoxic-preconditioned MSCs [123], minocycline-preconditioned NSCs [124], and electrically preconditioned human NPCs [125] reportedly improved functional recovery in acute or subacute phase transplantation after ischemic stroke, these preconditioning strategies could enhance stem cell function [126]. Optogenetically or optochemogenetically stimulated transplanted NSCs, with enhanced trophic factors and neurogenesis, could lead to functional recovery [127,128]. Rejuvenescence of stem cells might also improve stem cell function [129], and might be effective, especially in autologous stem cell transplantation. Bcl-2 is one of the anti-apoptotic genes, and lentiviral transfected Bcl-2 expressing human NSCs reportedly has better survival in the brain and improves functional recovery after ischemic stroke in acute phase transplantation [130]. Prolonged survival stem cells could also enhance the replacement effect. Thus, functionally enhanced stem cells could improve therapeutic efficacy.

### 8.2. Enhancement of Transplanted Stem Cell Migration to an Infarct Site

Mannitol is an established agent for improving the permeability of the BBB. Mannitol has been reported to enhance the delivery of stem cells and growth factors into the brain during transplantation [131]. Free radical scavengers increased the expression of SDF-1 leading to increased transplanted stem cell migration into the injury site [132]. Moreover, physical approaches, such as noninvasive microbubble-enhanced FUS, can safely and transiently increase BBB permeability at the target lesion. Oscillation of the intravenously injected microbubbles in vasculature exposed to ultrasound irradiation not only alters the permeability of the BBB but can also enhance the delivery of chemotherapeutics (Figure 5). FUS/microbubble-assisted BBB opening has been demonstrated in patients with Alzheimer’s disease and malignant glioma using MRI, in which regional contrast extravasation was confirmed [133,134,135]. Our group showed that FUS/microbubble-assisted BBB opening enhanced LNP-mediated mRNA delivery to the mouse brain [4]. More recently, cell delivery induced BBB opening via non-FUS was evaluated in a brain-ischemia rat model [136]. This study showed a significant increase in intravenously administered MSC engraftment with minor improvement in neurological outcomes over intravenous injection of the MSC alone. There are few animal studies on the combination of FUS/microbubble and MSC [137]. Although we have established the procedure (Figure 5b) [4], it is still in the preliminary stage. More animal studies are needed before clinical trials. Although the application of IA MSC transplantation for ischemic stroke needs further examination, it is believed that FUS/microbubble-assisted BBB opening will soon become an important approach in MSC therapy.

## 9. Conclusions

In this paper, we reviewed the mechanisms of, and recent advancements in, stem cell transplantation for ischemic stroke, focusing on acute/subacute phase IA transplantation from preclinical to clinical studies. In acute/subacute IA transplantation, the bystander effect plays a key role, and the replacement effect might have limited treatment efficacy. The bystander effect includes angiogenesis, neurogenesis, and anti-inflammatory effects via a paracrine effect by transplanted stem cells; these also synergistically enhance tissue repair, as shown above. Preclinical studies have proven the efficacy of stem cell therapy for ischemic stroke using IA transplantation. Although clinical studies have demonstrated the safety of IA transplantation, efficacy is yet to be established. Multiple confounding variables, including optimal timing, cell dose, cell type, stroke type, stroke severity, and patient age, would explain the discrepancy in treatment efficacy between preclinical and clinical studies. Therefore, optimization of IA stem cell transplantation is necessary, and then further experiments in basic research are needed to fill these discrepancies and confirm the optimal conditions for IA stem cell transplantation. In the clinical setting, clinical induction of gene-modified stem cells, which could enhance the paracrine effect and/or prolong the lifespan of stem cells in the brain, could contribute to the improvement of the bystander and replacement effects. Selection of cell types containing a high density of stem cells, instead of BMMCs; increase in the infused stem cell dose; and clinical induction of new BBB opening techniques, such as the FUS/microbubble-assisted BBB opening technique, could also be helpful in improving the therapeutic effect. Further examination in the clinical setting is also needed to improve therapeutic efficacy while avoiding severe adverse effects, especially when a new technique moves from a preclinical to a clinical study setting. Our in-depth review will promote the efficient design of experiments and treatments, which will lead to better patient management and outcomes.

## Figures and Tables

**Figure 1 bioengineering-10-00033-f001:**
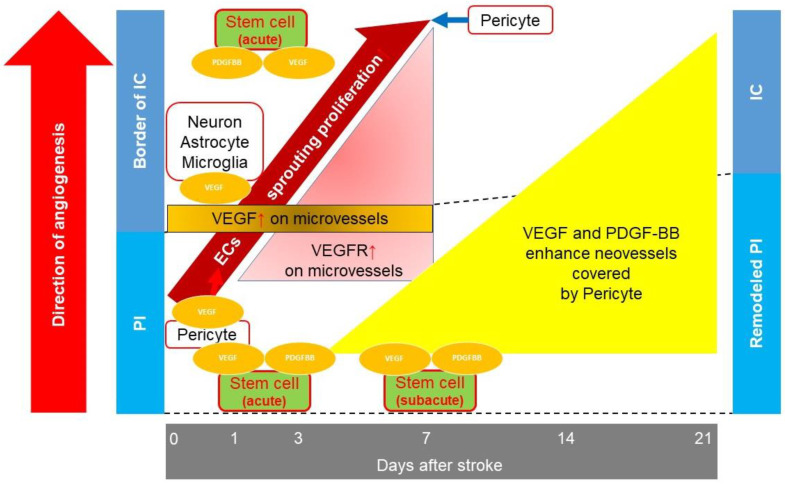
Summary of angiogenesis after intraarterial (IA) stem cell transplantation in acute and subacute phases post ischemic stroke. At the border of the infarct core (IC), within 24 h of ischemic stroke, neurons, astrocytes, and microglia secrete vascular endothelial growth factor (VEGF). The level of VEGF from microvessels increases after ischemic stroke, peaks on day 3, and normalizes until day 7 post ischemic stroke. In the periinfarct area (PI), pericytes secrete VEGF, helping endothelial cells (ECs) to proliferate. The expression of VEGF receptors (VEGFRs) on microvessels increases by day 2 post ischemic stroke, peaks on day 3, and normalizes until day 7. The expression of VEGFR and proliferation of ECs spreads from the PI to the IC. Once there is sufficient angiogenesis, pericytes terminate the sprouting proliferation of ECs. In the subacute phase, VEGF and platelet-derived growth factor-BB (PDGF-BB) enhance mature angiogenesis. In acute phase IA stem cell transplantation, stem cells are widely distributed in the IC and PI. In subacute phase IA stem cell transplantation, stem cells are mainly distributed in the PI. Transplanted stem cells secrete various growth factors, including VEGF and PDGF-BB, to promote angiogenesis. Thus, stem cell transplantation can reduce the infarct volume (red arrow, increase or enhancement; blue arrow, decrease or inhibition).

**Figure 2 bioengineering-10-00033-f002:**
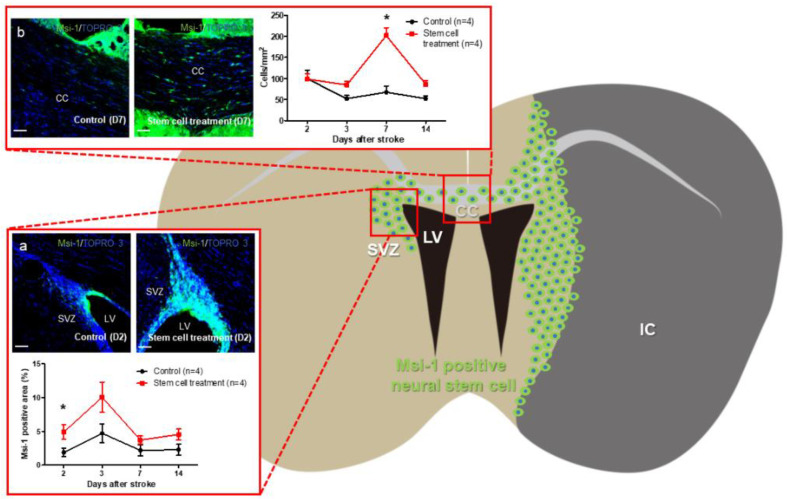
Sprague-Dawley rats were anesthetized with isoflurane, and a 75-min transient middle cerebral artery occlusion (MCAO) was performed with a 4.0 nylon monofilament suture coated with silicone (4039PK10; Doccol). One day after MCAO, the rats underwent stem cell transplantation. Mesenchymal stem cells (1 × 10^6^ cells) or phosphate-buffered saline were injected into the internal carotid artery through a catheter inserted from the stump of the distal external carotid artery to the internal carotid artery through the bifurcation. After deep anesthesia, the brain was excised following perfusion fixation using 4% paraformaldehyde. After adequate fixation, a 40-μm frozen coronal section was made [11]. Three coronal sections of the brain (1.0 ± 0.6 mm before the bregma) were stained using Musashi-1 (rat anti-Musashi-1 [Msi-1]: Medical and Biological Laboratories Co., Ltd., Nagoya, Japan; 1:100). The area of Msi-1-positive neural stem cells was measured using WinRoof software (Mitani Corporation; Tokyo, Japan) in the subventricular zone (SVZ). The Msi-1-positive neural stem cells were measured by counting Msi-1- and TOPRO-3- double positive cells in the corpus callosum (CC). (**a**) The Msi-1-positive area in the SVZ of the contralateral hemisphere in the stem cell treatment group was significantly larger than that in the control group 2 days after MCAO. (**b**) Seven days after MCAO, the number of Msi-1-positive cells in the CC in the stem cell treatment group was significantly higher than that in the control group. All data are presented as means ± SEM * *p* < 0.05, (student *t*-test). IC; infarct core: LV; lateral ventricle.

**Figure 3 bioengineering-10-00033-f003:**
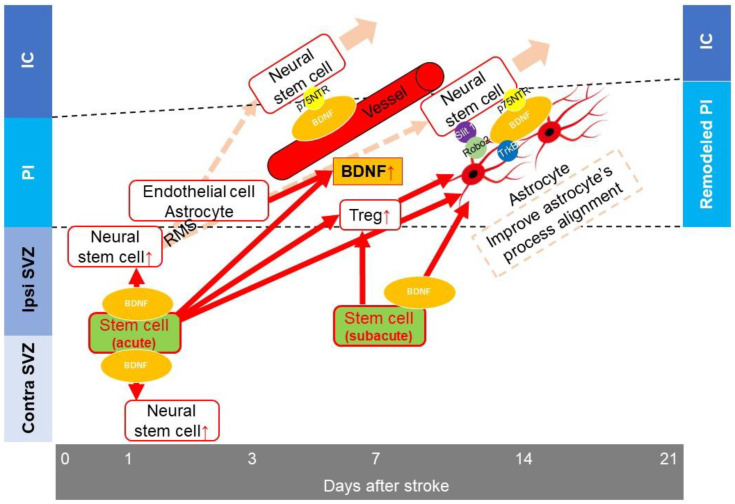
Neurogenesis. After ischemic stroke, the proliferation of neural stem cells in the subventricular zone (SVZ) increases. These cells migrate to the injury site via the rostral migratory stream (RMS). Astrocytes and endothelial cells secrete brain-derived neurotrophic factor (BDNF). Three to fifteen days post ischemic stroke, neural stem cells (NSCs) migrate into the periinfarct area along the vessels, with p75NTR on the NSCs binding to BDNF on the vessels. Seven to fifteen days post ischemic stroke, the cells bind to BDNF on astrocytes. NSCs also change the morphology of astrocytes through Slit-1-Robo signaling and migrate into the periinfarct area along astrocytes. Regulatory T cells (Tregs) also control neurotoxic astrogliosis. Intraarterial (IA) transplantation provides stem cells in both hemispheres. Acute phase IA transplantation enhances neural stem cell proliferation in both the ipsilateral and contralateral SVZ and increases the level of BDNF in the periinfarct area 7 days post ischemic stroke. These events enhance neural stem cell migration. In addition, transplanted stem cells also enhance the proliferation of Tregs. IA transplantation could enhance NSC migration to the periinfarct area by control of astrogliosis. Three days post ischemic stroke, the ischemic pericytes express NSC markers (red arrow, increase or enhancement). IC: infarct core; PI: peri-infarct lesion: ipsi: ipsilateral; contra: contralateral.

**Figure 4 bioengineering-10-00033-f004:**
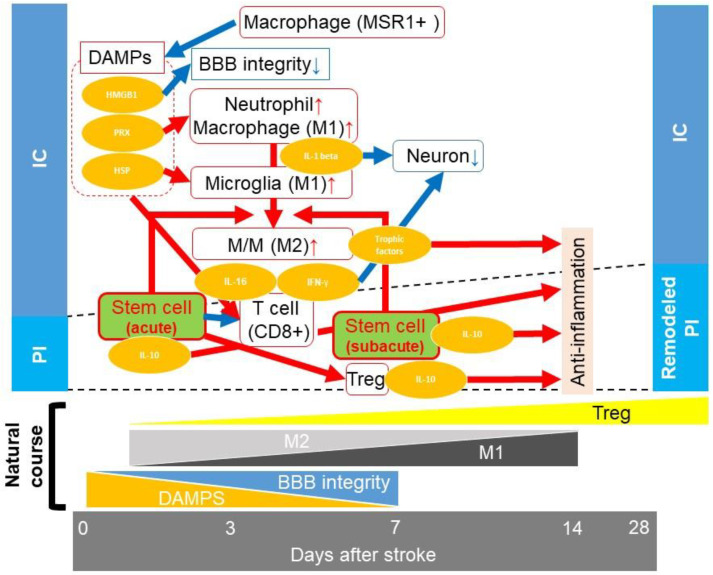
Summary of anti-inflammatory processes. After an ischemic stroke, the level of damage-associated molecular patterns (DAMPs) increases. DAMPs activate macrophages/microglia (M/M) (M1) and neutrophils. DAMPs also increase vessel permeability by decreasing blood-brain barrier (BBB) integrity, leading to infiltration of inflammatory cells into the infarct area. M/M (M1) secretes interleukin (IL)-1 beta, which injures neurons directly. From 3 days post ischemic stroke, MSR1-expressing macrophages, which scavenge DAMPS, increase up to 7 days post ischemic stroke. As DAMPs start decreasing, BBB integrity improves and neuroinflammation decreases. A few days post ischemic stroke, T cells infiltrate the infarct lesion. CD8+ T cells produce interferon-γ and IL-16, injuring neurons. Regulatory T cells (Tregs) gradually increase from the acute phase to 60 days post ischemic stroke. Tregs produce IL-10, reducing the infarct volume. In acute and subacute phase IA stem cell transplantation, stem cells switch the M/M polarization from M1 to M2, which reduces debris and secretes various trophic/growth factors. Stem cells also secrete IL-10. These effects contribute to reducing the infarct volume (red arrow, increase or enhancement; blue arrow, decrease or inhibition). IC: infarct core; PI: peri-infarct lesion.

**Figure 5 bioengineering-10-00033-f005:**
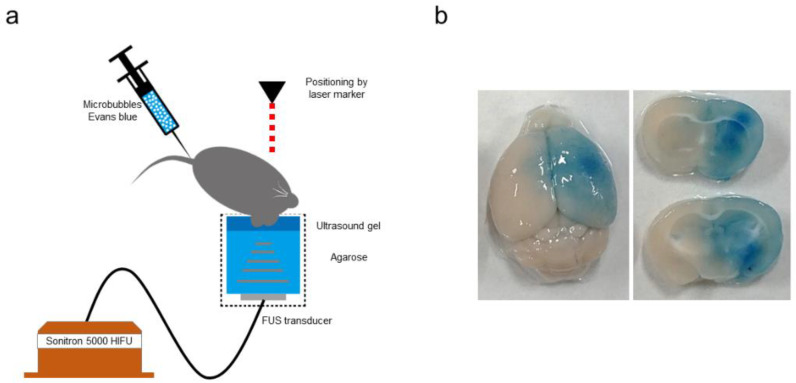
Focused ultrasound (FUS)/microbubble-assisted blood-brain barrier (BBB) opening system. (**a**) Schematic of the experimental setup for BBB opening. The cylindrical FUS transducer was filled with agarose and ultrasound gel. The mouse was placed in the supine position, and the head positioned on ultrasound gel. The irradiation target region in the mouse brain was adjusted using a laser marking device. Then, FUS (frequency, 3 MHz; intensity, 1.5 kW/cm^2^; duration, 60 s; duty cycle, 5% [1 msec irradiation and 19 msec interval]) was applied to the right striatum after intravenous administration of microbubbles and Evans blue. (**b**) BBB opening confirmed using Evans blue extravasation.

**Table 1 bioengineering-10-00033-t001:** Clinical studies of IA transplantation.

Reference	Cell Type	Cell Source	Cell Dose/Total Volume	Infusion Speed	Infarct Area	Infusion Site	Timing of Transplantation Mean (Min–Max)	No. of Treated Patients /No. of Controls	Adverse Effects (Procedure-Related and Delayed Complication)	Findings
Correa, P. L. 2007	Autologous	BMMC	3.0 × 10^7^/NR	NR	MCA	MCA	9 days	1/0	None	NR
Friedrich, M. A. 2012	Autologous	BMMC	Mean 22.08 × 10^7^ (5.1–60 × 10^7^) /15 mL	0.5 mL/min	MCA	MCA	6 ± 1.8 days (3–10)	20/0	None	There was significant reduction in the median pretreatment NIHSS score over the 180 days post transplantation.
Moniche, F. et al., 2012	Autologous	BMMC	Mean 1.59 × 10^8^/NR	0.5–1.0 mL/min	MCA	M1	6.4 ± 1.3 (5–9 days)	10/10	No procedure-related adverse effects. Two seizures.	No significant differences in neurological function between the treatment and control groups. The better BI at one month correlated with a higher number of CD34+ cells.
Jiang, Y. et al., 2013	Allogenic	UCMSC	2.0 × 10^7^/20 mL	1 mL/min	MCA	M1	17.3 ± 5.7 (11–19 days)	3/0	None	Improvement in mRS was seen in 2/3 patients at 90 and 180 days.
Rosado-de-Castro, P. H. et al., 2013	Autologous	BMMC	Mean 2.80 × 10^8^ (1.25–5×10^8^)/10 mL	1 mL/min	MCA	MCA	Mean 62 ± 20.4 (19–82 days)	7/0	No procedure-related adverse effects. Two had seizures.	No patients with worsened scores of BI, mRS, or NIHSS.
Banerjee, S. et al., 2014	Autologous	CD34+ stem cells	Mean 2.2 × 10^6^ (1.2–2.79 × 10^6^)/NR	NR/10 min	MCA	MCA	Within 7 days	5/0	No procedure-related adverse effects. One had renal dysfunction.	There was a significant difference in mean NIHSS and mRS before and 180 days after transplantation. No reduction of the mean infarct volume. No new lesions on MRI.
Ghali, A. A. 2016	Autologous	BMMC	About 1 × 10^6^/100 mL	NR	MCA	ICA	Mean 22 (12–32 days)	21/18	None	There were no significant differences in the improvement of NIHSS and BI between the stem cell and control groups at 12 months.
Bhatia, V. et al., 2018	Autologous	BMMC	Mean 6.1 × 10^8^ (maximum 5 × 10^8^)/mean 5 mL	Mean 0.5 mL/min	MCA	M1	Mean 10 (8–15 days)	10/10	No procedure-related adverse effects.	Compared with controls, there was increase in the incidence of good outcomes.
Savitz, S. I. et al., 2019	Autologous	BMMC	Mean 3.08 × 10^6^ (1.6 × 10^5^–7.5 × 10^7^)/2.7 ± 0.8 mL	2.7 ± 0.8 mL/2 to 3 min	MCA	ICA	13–19 days	29/19	Four asymptomatic infarctions.	There was no significant efficacy.

BMMC: bone marrow mononuclear cell; UCMSC: umbilical cord mesenchymal stem cell; MCA: middle cerebral artery; ICA: internal carotid artery; BI: Barthel index; NIHSS: National Institutes of Health Stroke Scale; mRS: modified Rankin Scale; NR: not recorded.

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
