# Peer review of "Stem Cell Therapy for Acute/Subacute Ischemic Stroke with a Focus on Intraarterial Stem Cell Transplantation: From Basic Research to Clinical Trials"

_bioengineering, 2022, doi:10.3390/bioengineering10010033_

Round 1

Reviewer 1 Report (Previous Reviewer 1)

The title of the paper is intraarterial cell transplant - focusing on that may help make the paper more manageable.  Alternatively broaden scope to be cellular tx for acute/subacute stroke.  You struggle with covering both areas of review

The added information is factually correct but the wording/context needs help.  I can see where you are coming from.

Author Response

Thank you for giving us the opportunity to submit a revised draft of our manuscript titled “Stem cell therapy for acute/subacute ischemic stroke with a focus on intraarterial stem cell transplantation: From basic research to clinical trials” to Bioengineering. We appreciate the time and effort that you and the reviewers have dedicated to providing your valuable feedback on my manuscript. We are grateful to the reviewers for their insightful comments on our paper. We have been able to incorporate changes to reflect most of the suggestions provided by the reviewers. We have highlighted the changes within the manuscript in red-colored font. Here is a point-by-point response to the reviewers’ comments and concerns.

Reviewer 1:

The title of the paper is intraarterial cell transplant - focusing on that may help make the paper more manageable.  Alternatively broaden scope to be cellular tx for acute/subacute stroke.  You struggle with covering both areas of review.

Response: We thank the reviewer for the helpful comments. According to this beneficial suggestion, we have changed the title that better represents our study (Lines 2–4).

The added information is factually correct but the wording/context needs help.  I can see where you are coming from.

Response: We thank the reviewer for pointing this out. We have submitted our manuscript to an English editing company (Editage) to proofread and recheck the manuscript for correct language, word usage, and context (Lines 8792, 100–104, 137–144, 146–166, 168–169, 173–182, 224–229, 446–466, 503-508, 514–519, 557–563 and 576–577).

Reviewer 2 Report (New Reviewer)

It should be published in its present form. 

Author Response

Thank you for giving us the opportunity to submit a revised draft of our manuscript titled “Stem cell therapy for acute/subacute ischemic stroke with a focus on intraarterial stem cell transplantation: From basic research to clinical trials” to Bioengineering. We appreciate the time and effort that you and the reviewers have dedicated to providing your valuable feedback on my manuscript. We are grateful to the reviewers for their insightful comments on our paper. We have been able to incorporate changes to reflect most of the suggestions provided by the reviewers. We have highlighted the changes within the manuscript in red-colored font. Here is a point-by-point response to the reviewers’ comments and concerns.

Reviewer 2

It should be published in its present form. 

Response: We thank the reviewer for the kind evaluation of our manuscript.

Reviewer 3 Report (New Reviewer)

This review paper summarized the mechanism and current research progress of stem cell transplantation in ischemic stroke treatment. It's a very comprehensive review and the structure of the paper is well constructed. Below are some suggestions which may further improve the quality of the article and facilitate the work of researchers in related field.

1. The authors emphasized that the key to stem cell transplantation for ischemic stroke is to improve the therapeutic effect. What is the stem cell distribution efficiency in brain and spleen after IA transplantation? If the author gave some details of the stem cell distribution efficiency, it may be helpful for researchers in the field.

2. What is the residence time of stem cell after transplantation? The related technique to identify the properties of stem cell after transplantation might be emphasized.

3. What is the advantages of stem cell transplantation for ischemic stroke as compared to the traditional drug? If possible, some comparation between current therapeutic agents with stem cell transplantation may be interesting, even in the preclinical study.

Author Response

Thank you for giving us the opportunity to submit a revised draft of our manuscript titled “Stem cell therapy for acute/subacute ischemic stroke with a focus on intraarterial stem cell transplantation: From basic research to clinical trials” to Bioengineering. We appreciate the time and effort that you and the reviewers have dedicated to providing your valuable feedback on my manuscript. We are grateful to the reviewers for their insightful comments on our paper. We have been able to incorporate changes to reflect most of the suggestions provided by the reviewers. We have highlighted the changes within the manuscript in red-colored font. Here is a point-by-point response to the reviewers’ comments and concerns.

Reviewer 3

This review paper summarized the mechanism and current research progress of stem cell transplantation in ischemic stroke treatment. It's a very comprehensive review and the structure of the paper is well constructed. Below are some suggestions which may further improve the quality of the article and facilitate the work of researchers in related field.

1.The authors emphasized that the key to stem cell transplantation for ischemic stroke is to improve the therapeutic effect. What is the stem cell distribution efficiency in brain and spleen after IA transplantation? If the author gave some details of the stem cell distribution efficiency, it may be helpful for researchers in the field.

Response: We thank the reviewer for this pertinent comment. As mentioned in the manuscript, the location of transplanted stem cells in the brain can affect the angiogenesis and neurogenesis (Lines 220-224 and 291-293) and the timing of transplantation can affect the distribution of transplanted stem cells in the brain (Lines 119-128). The main role of transplanted stem cells in the spleen is antiinflammation. Therefore, timing of transplantation is also be important in systemic effect. We have added a sentence regarding the distribution of stem cells in the liver and spleen (Lines 454-457).

  1. What is the residence time of stem cell after transplantation? The related technique to identify the properties of stem cell after transplantation might be emphasized.

Response: We thank the reviewer for the helpful comments. We have added the information according to the reviewer’s request. (Lines 473-479).

  1. What is the advantages of stem cell transplantation for ischemic stroke as compared to the traditional drug? If possible, some comparation between current therapeutic agents with stem cell transplantation may be interesting, even in the preclinical study.

Response: We thank the reviewer for this helpful comment. We could not find a comparative clinical study on the therapeutic benefits of stem cells and traditional drugs for ischemic stroke. However, we found a preclinical study using free radical scavenger-treated rat models. Free radical scavenger is one of the neuroprotective agents that plays a role after ischemic stroke. One report showed that there were no significant differences in functional recovery between administration of free scavenger and BMSC. However, a combination of BMSC and free radical scavenger improved functional recovery in rats compared with rats treated with the free radical scavenger alone or those treated with BMSC alone. This effect was due to the free radical scavenger-induced migration of transplanted BMSCs via increased levels of SDF-1 expression in ischemic brain (Shen et al. Chin Med J 2016). Free radical scavenger also improved the environment of the ischemic brain, and this improved the survival of transplanted BMSCs (Shen et al. Neuroscience 2012). We have added this information on Lines 484-490 and 612-613 of the revised manuscript.

This manuscript is a resubmission of an earlier submission. The following is a list of the peer review reports and author responses from that submission.

Round 1

Reviewer 1 Report

This is a focused review of the experience to date with intra-arterial infusion of cells for treatment of ischemic stroke - both literature review and update of the author's experience.  The review covers preclinical and clinical reports.  The review is thorough.  Highlights the safety of infusing cells and the mixed reports from small series - ranging from no to some response in small series.

The review highlights areas that need attention with intraarterial cell delivery to cell dose, avoiding embolism, timing after the stroke, etc. Limited comments on the technical challenges (volume, rates of infusions, where the catheter is placed wrt infarct, timing, etc).  

The review superficially addresses the various cell populations that have be infused - mesenchymal stem/stromal cells, cells with neuronal commitment from a range of sources, hematopoietic progenitors.  This is especially important given that all the studies summarized in Table 1 were hematopoietic stem/progenitor cells and not the mesenchymal stem/progenitor cells discussed in the author's own work

Specific comments:

lines 368-9: 'Seizure seemed to be an adverse event which was not associated with the procedure but was related to IA transplantation' is confusing -- delivering cells is part of the procedure.  Perhaps a clarification

Author Response

Response to editors

We thank the editor and reviewers for their comments and feedback. We have had the manuscript re-edited for language, and our responses to all the comments are provided below.

Reviewer 1

This is a focused review of the experience to date with intra-arterial infusion of cells for treatment of ischemic stroke - both literature review and update of the author's experience. The review covers preclinical and clinical reports. The review is thorough. Highlights the safety of infusing cells and the mixed reports from small series - ranging from no to some response in small series.

The review highlights areas that need attention with intraarterial cell delivery to cell dose, avoiding embolism, timing after the stroke, etc. Limited comments on the technical challenges (volume, rates of infusions, where the catheter is placed wrt infarct, timing, etc).

The review superficially addresses the various cell populations that have be infused - mesenchymal stem/stromal cells, cells with neuronal commitment from a range of sources, hematopoietic progenitors. This is especially important given that all the studies summarized in Table 1 were hematopoietic stem/progenitor cells and not the mesenchymal stem/progenitor cells discussed in the author's own work

Response: Thank you for your comments. We have added sections elaborating on acute and subacute phase MSC transplantation. (Lines 475-478)

Specific comments:

lines 368-9: 'Seizure seemed to be an adverse event which was not associated with the procedure but was related to IA transplantation' is confusing -- delivering cells is part of the procedure. Perhaps a clarification

Response: Thank you for your comment. We agree that this sentence could confuse reader, and have rephrased it for clarity. (Lines 433-435)

Reviewer 2 Report

Title: Acute/subacute phase intraarterial stem cell transplantation for ischemic stroke: From basic research to clinical trial

Journal: Bioengineering

The topic is of interest, and the manuscript is well illustrated.

Major Comments:

1. Are there controversies in this field? What are the most recent and important achievements in the field? In my opinion, answers to these questions should be emphasized. Perhaps, in some cases, novelty of the recent achievements should be highlighted by indicating the year of publication in the text of the manuscript.

2. Mechanisms of stem cell transplantation for ischemic stroke: This section is very weak and no emphasis is given.

3. Cell replacement is also very weak.

4. Conclusion: The section devoted to the explanation of the results suffers from the same problems revealed so far. Your storyline in the results section (and conclusion) is hard to follow. Moreover, the conclusions reached are really far from what one can infer from the empirical results.

5. The overall discussion should be rather organized around arguments avoiding simply describing details without providing much meaning. A real discussion should also link the findings of the study to theory and/or literature.

6. Spacing, punctuation marks, grammar, and spelling errors should be reviewed thoroughly. I found so many typos throughout the manuscript.

7. English is modest. Therefore, the authors need to improve their writing style. In addition, the whole manuscript needs to be checked by native English speakers.

Author Response

Response to editors

We thank the editor and reviewers for their comments and feedback. We have had the manuscript re-edited for language, and our responses to all the comments are provided below.

Reviewer 2

Major Comments:

  1. Are there controversies in this field? What are the most recent and important achievements in the field? In my opinion, answers to these questions should be emphasized. Perhaps, in some cases, novelty of the recent achievements should be highlighted by indicating the year of publication in the text of the manuscript.

Response: Thank you for your helpful comments. Accordingly, we have added a statement about the recent advancements in drug and stem cell therapy. (Lines 47-61, 64-69)

  1. Mechanisms of stem cell transplantation for ischemic stroke: This section is very weak and no emphasis is given.

Response: Thank you for your helpful comments. The main role of acute and subacute phase IA stem cell transplantation is the bystander effect, which can be enhanced by gene-modified stem cell transplantation. We have added information on gene-modified stem cell transplantation to our manuscript. (Lines 374-391, 501-503).

  1. Cell replacement is also very weak.

Response: Thank you for helpful comments. Since the main role of acute and subacute phase IA transplantation is the bystander effect, the replacement effect is limited. We have added a discussion regarding the cell replacement effect. (Lines 403-411 and 418-422)

  1. Conclusion: The section devoted to the explanation of the results suffers from the same problems revealed so far. Your storyline in the results section (and conclusion) is hard to follow. Moreover, the conclusions reached are really far from what one can infer from the empirical results.

Response: Thank you for your useful advice, and we apologize for any troubles in reviewing our manuscript. Accordingly, we have rewritten and added a significant amount of information to our manuscript. We have now explained the mechanism related to cerebral infarction and cell transplantation in detail. Although there was much crosstalk among each effect (e.g., between angiogenesis and neurogenesis, antiinflammation and angiogenesis), we divided the mechanism into 3 main sections to increase improve the readability of our manuscript. To clarify the crosstalk provided by stem cell transplantation, we also added the one section and summarized crosstalk between each mechanism of stem cell therapy (Lines 393-400).

We rephrased the following sentence to improve the readability (Lines 177-184)

We also added this sentence to our conclusion (Lines 545-546 and 550-552)

  1. The overall discussion should be rather organized around arguments avoiding simply describing details without providing much meaning. A real discussion should also link the findings of the study to theory and/or literature.

Response: Thank you for your helpful comments. We have linked the findings to literature (Line 249) and added a discussion to each section (Lines 393-400, 418-422, 447-456, 467-469, 475-484, 486-489)

  1. Spacing, punctuation marks, grammar, and spelling errors should be reviewed thoroughly. I found so many typos throughout the manuscript.

Response: Thank you for your comment. We have revised the manuscript carefully.

  1. English is modest. Therefore, the authors need to improve their writing style. In addition, the whole manuscript needs to be checked by native English speakers.

Response: Thank you for your feedback. Although a native speaker from Editage edited the original manuscript, I have asked them to carefully edit the revised manuscript as well.

Reviewer 3 Report

 In the present review, the authors discussed the usefulness of intraarterial stem cell transplantation for ischemic stroke at acute/subacute phase. Although the description is comprehensive and informative, more detailed presentation is required.

Major comments

1. Outcomes in clinical trials using intraarterial stem cell transplantation is not impressive in spite of its efficacy in animal models.  The reason for this discrepancy should be more clearly demonstrated, showing the exact data. 

2. The authors should show more precisely the difference between the outcome of intraarterial stem cell transplantation and that of intravenous stem cell transplantation in clinical settings.   

Author Response

Response to editors

We thank the editor and reviewers for their comments and feedback. We have had the manuscript re-edited for language, and our responses to all the comments are provided below.

Reviewer 3

 In the present review, the authors discussed the usefulness of intraarterial stem cell transplantation for ischemic stroke at acute/subacute phase. Although the description is comprehensive and informative, more detailed presentation is required.

Major comments

  1. Outcomes in clinical trials using intraarterial stem cell transplantation is not impressive in spite of its efficacy in animal models. The reason for this discrepancy should be more clearly demonstrated, showing the exact data.

Response: Thank you for your useful suggestion. We have added a statement regarding this discrepancy between preclinical studies using animal models and clinical trials (Lines 467-469, 475-484and 486-489).

  1. The authors should show more precisely the difference between the outcome of intraarterial stem cell transplantation and that of intravenous stem cell transplantation in clinical settings.

Response: Thank you for this helpful advice. Accordingly, we have added the findings of IV transplantation in clinical trials and discussed the different outcomes of IV and IA (Lines 447-456)

Round 2

Reviewer 1 Report

The modifications to the manuscript unfortunately do not add clarity.  I believe the authors are exploring the issues around intraarterial vs other routes of administration - the risks and potential benefits.  

This is a difficult task given the many confounding variables in the preclinical and clinical experience to date - type and timing of CNS insult prior to cell infusion, type of cells infused, whether these cells remain in the circulation/tissue long enough to provide a biologic effect (most likely paracrine).  Most studies are not able to measure the cells after infusion.

These are important questions that the review could put into a framework that would support future thinking/research.  My apologies if I am not comprehending the goal of the manuscript.

Author Response

Response to editors

We thank the editor and reviewers for their comments and feedback. We have had the manuscript re-edited for language, and our responses to all the comments are provided below.

Reviewer 1

The modifications to the manuscript unfortunately do not add clarity.  I believe the authors are exploring the issues around intraarterial vs other routes of administration - the risks and potential benefits.  

This is a difficult task given the many confounding variables in the preclinical and clinical experience to date - type and timing of CNS insult prior to cell infusion, type of cells infused, whether these cells remain in the circulation/tissue long enough to provide a biologic effect (most likely paracrine).  Most studies are not able to measure the cells after infusion.

These are important questions that the review could put into a framework that would support future thinking/research.  My apologies if I am not comprehending the goal of the manuscript.

Response: I agree with your comment! There are many confounding variables, and these variables make it difficult to find the optimal condition of stem cell transplantation, evaluate the therapeutic efficacy, and understand the underlying problems in failure of clinical trials. I hope this review can help many other researchers to advance stem cell therapy. I have changed the conclusion in reference to your comments (Lines 551 to 575).

Reviewer 2 Report

basic research to clinical trial

Journal: Bioengineering

The topic is of interest, and the manuscript is well illustrated.

Minor Comments:

1. Conclusion: The section devoted to the explanation of the results suffers from the same problems revealed so far. Your storyline in the results section (and conclusion) is hard to follow. Moreover, the conclusions reached are really far from what one can infer from the empirical results.

2. English is modest. Therefore, the authors need to improve their writing style. In addition, the whole manuscript needs to be checked by native English speakers.

Author Response

Response to editors

We thank the editor and reviewers for their comments and feedback. We have had the manuscript re-edited for language, and our responses to all the comments are provided below.

Reviewer 2

Basic research to clinical trial

Journal: Bioengineering

The topic is of interest, and the manuscript is well illustrated.

Minor Comments:

  1. Conclusion: The section devoted to the explanation of the results suffers from the same problems revealed so far. Your storyline in the results section (and conclusion) is hard to follow. Moreover, the conclusions reached are really far from what one can infer from the empirical results.

   Response: Thank you for your comments. I changed the conclusion to better link our results (551 to 575). I hope this change can help to increase the readability for readers.

  1. English is modest. Therefore, the authors need to improve their writing style. In addition, the whole manuscript needs to be checked by native English speakers.

Response: Thank you for your comments. I will do my best to resolve the English language issues of our manuscript with Editage, which is an English editing company and responsible for editing our manuscript concerning English issues.

Reviewer 3 Report

 The revised manuscript is much better than the previous one.

Author Response

Response to editors

We thank the editor and reviewers for their comments and feedback. We have had the manuscript re-edited for language, and our responses to all the comments are provided below.

Reviewer 3

The revised manuscript is much better than the previous one.

Response: Thank you for your contribution towards improving our manuscript.
